# Embedding Domain-Specific Invariances into Contrastive Learning for Calibration-Free Neural Decoding

## Abstract

Steady-state visual evoked potentials (SSVEPs) provide a high-throughput testbed for neural decoding, yet real-world deployment is hindered by subject-specific calibration. We address this challenge by proposing DATCAN, a framework that embeds domain-specific invariances into contrastive learning while aligning feature statistics without supervision. DATCAN integrates three complementary components: (i) a harmonic-aware contrastive objective that encodes frequency-locked physiological priors directly into the embedding space, (ii) second-order covariance alignment(CORAL) that stabilizes cross-subject transfer through closed-form adaptation, and (iii) adaptive late fusion of interpretable classical heads (Task-Related Component Analysis, TRCA; Filter-Bank Canonical Correlation Analysis, FBCCA)(Nakanishi et al., 2018; Chen et al., 2015) with normalized weighting.Contrastive pairing uses only source-subject labels: positives are other-subject trials evoked by the same known stimulus frequency (including harmonics), while negatives come from different frequencies. At inference, the TRCA/F-BCCA heads score each frequency class, mapping embeddings to symbols without any target-subject calibration. Evaluated under strict leave-one-subject-out transfer, DATCAN achieves robust short-window decoding, sustaining >100 bits/min information transfer rate at 1 s—a regime where **existing calibration-free baselines** substantially underperform. Ablation and interpretability analyses confirm that each module contributes principled gains, yielding physiologically grounded, subject-invariant representations. Beyond Electroencephalogram(EEG), our results highlight a general recipe for calibration-free domain adaptation: encode physics-driven invariances in contrastive learning, align covariances without labels, and integrate interpretable ensembles. This blueprint extends naturally to other sequential and biosignal domains where distribution shift and data scarcity remain central obstacles.

Reproducibility: Code, preprocessing scripts, and evaluation notebooks with fixed seeds are provided in the supplementary material (anonymous).

## 1 Introduction

**Calibration bottleneck.** Brain–computer interfaces (BCIs) (Wolpaw et al., 2002) promise assistive communication, yet per-subject calibration remains a barrier. In steady-state visual evoked potentials (SSVEPs)—the workhorse for high-throughput BCIs—*cross-subject* performance deteriorates in short windows (1–2 s), precisely where fast interaction is needed.

**Domain-adaptation view.** Across subjects, electrode placement, anatomy, and noise induce *domain shift*, while stimulus-locked harmonics are preserved. The objective is to learn *harmonic-invariant, subject-agnostic* representations without target labels. Classical TRCA/FBCCA pipelines (Nakanishi et al., 2018; Chen et al., 2015) leverage reproducibility and harmonic priors but falter at 1–2 s; deep models often require fine-tuning.

**Our approach: DATCAN.** DATCAN is a calibration-free framework that encodes physics into contrastive learning, reduces subject shift with lightweight statistics, and fuses interpretable heads at decision time: (i) *harmonic-conditioned InfoNCE* treats cross-subject trials sharing stimulus frequency (and harmonics) as positives; (ii) *CORAL* performs closed-form second-order alignment without target labels or adversarial training; and (iii) *adaptive late fusion* z-normalizes and combines TRCA (reproducibility) with FBCCA (harmonic specificity).

**Calibration-free 1.0 s benchmark.** Under strict LOSO with no target calibration/adaptation, prior methods typically fall below ∼100 bits/min, whereas calibration- or adaptation-based approaches (e.g., msSAME (Luo et al., 2023), SFDA-SSVEP (Guney et al., 2023)) report higher ITRs. DATCAN achieves **141.4 bits/min** (best subject) and ≈ 100+ **bits/min** on average at 1.0 s in a fully calibration-free setting.

**Contributions.**
- **Physics-guided SSL for EEG:** a harmonic-conditioned contrastive loss that yields frequency-aligned, subject-invariant embeddings in short windows.
- **Label-free alignment:** CORAL on dual-head embeddings provides stable cross-subject transfer via a simple closed-form transform.
- **Interpretable, robust decoding:** an adaptive TRCA+FBCCA ensemble sustains calibration-free performance on two multi-subject benchmarks, exceeding 100 bits/min at 1.0 s.

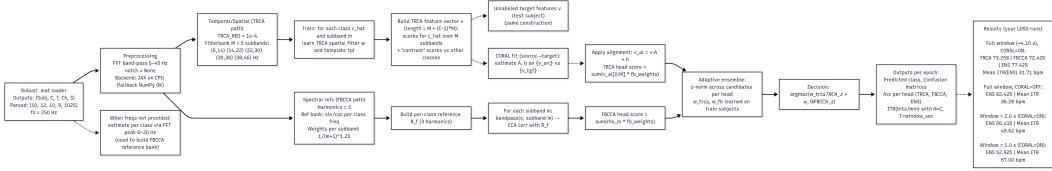

Figure 1: *DATCAN overview.* Filter-bank EEG → dual heads (TRCA, FBCCA) → harmonic-aware contrastive embeddings + CORAL alignment → adaptive fusion. The design preserves interpretability while enabling calibration-free LOSO decoding at 1.0 s.

## 2 RELATED WORK

**Classical SSVEP decoding.** CCA/FBCCA correlate EEG with sinusoidal references (incl. harmonics), while TRCA learns spatial filters maximizing trial-to-trial reproducibility (Chen et al., 2015; Nakanishi et al., 2018). These pipelines are efficient and interpretable but degrade under *cross-subject* transfer and *short windows* ($< 2$ s) due to subject-specific covariance shift. **DATCAN** retains their harmonic priors and transparency while adding learned invariances and alignment tailored for transfer.

**Deep and hybrid EEG decoders.** Compact CNNs (e.g., EEGNet) (Lawhern et al., 2018) can match/exceed handcrafted methods in within-subject settings, whereas cross-subject generalization typically needs calibration or fine-tuning. Hybrid filter-bank + CNN designs raise capacity but often trade off latency, interpretability, or calibration-free transfer. **DATCAN** keeps linear-time classical heads, using representation learning only where subject shift arises.

**Domain adaptation and self-supervision.** Adversarial alignment (e.g., gradient reversal) (Ganin et al., 2016) can reduce domain gaps but is brittle for low-SNR EEG. CORAL (Sun et al., 2016; Sun & Saenko, 2016) aligns second-order statistics via a closed-form linear map, avoiding adversarial instability and extra labels. Generic contrastive SSL (Chen et al., 2020; van den Oord et al., 2018) reduces label reliance but ignores SSVEP physics (harmonics, phase locking). **DATCAN** combines: (i) *harmonic-conditioned InfoNCE* treating cross-subject trials at the same stimulus frequency (and harmonics) as positives, and (ii) *CORAL* on dual-head embeddings, yielding label-free adaptation with lightweight inference.

**Positioning.** Prior ingredients—FBCCA/TRCA, CORAL, and SSL—are partial. A unified, *calibration-free* pipeline that (i) encodes harmonic priors in the embedding, (ii) performs closed-form target covariance alignment, and (iii) adaptively fuses complementary classical heads has been missing. **DATCAN** closes this gap, sustaining 1.0 s LOSO transfer with practical throughput (e.g., $> 100$ bits/min) without per-subject calibration.

## 3 METHOD: DATCAN FRAMEWORK

**DATCAN** is a calibration-free SSVEP pipeline that (i) learns frequency-invariant embeddings via a *harmonic-conditioned* InfoNCE objective using only source-subject labels, (ii) reduces inter-subject covariance shift with label-free, closed-form CORAL in embedding space, and (iii) *adaptively late-fuses* two interpretable heads—TRCA (trial-to-trial reproducibility) and FBCCA (harmonic

Table 1: ITR for calibration-reduced / fully calibration-free methods at $\sim 1.0\,$s windows.

| Method | Calibration / Adaptation | ITR (bits/min) |
|---|---|---|
| msSAME Luo et al. (2023) | minimal calibration ($\sim 24\,$s) | 213.8 |
| SFDA-SSVEP Guney et al. (2023) | unlabeled adaptation | 201.15 / 145.02 |
| iFuzzyTL ? | transfer learning | 213.99 / 94.63 |
| **Ours (DATCAN)** | fully calibration-free (LOSO) | 141.4 / $\approx 100+$ |

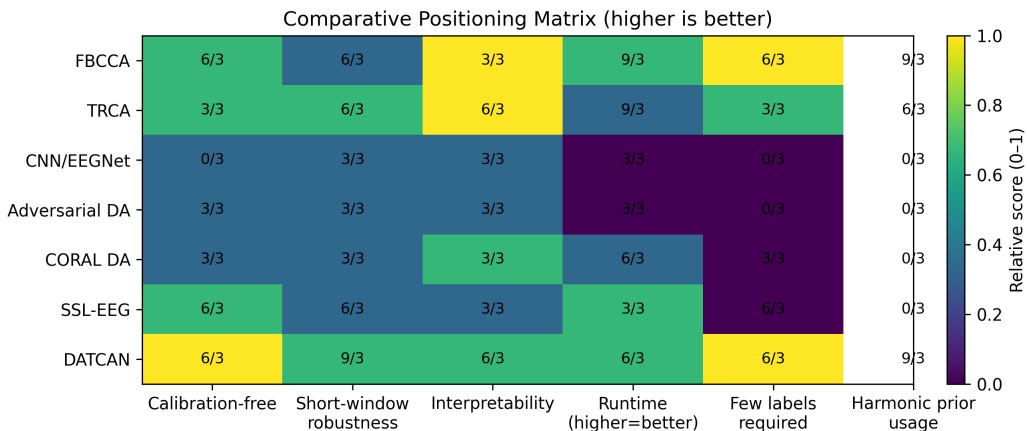

Figure 2: *Positioning.* Classical (FBCCA/TRCA) are interpretable but weaken under cross-subject transfer; deep CNNs typically need calibration. **DATCAN** integrates harmonic-aware SSL, CORAL alignment, and adaptive TRCA/FBCCA fusion to preserve interpretability and achieve robust 1 s decoding.

specificity). At test time, head scores map embeddings $\rightarrow$ symbols with *no* target calibration, sustaining short-window (1.0 s) LOSO decoding.

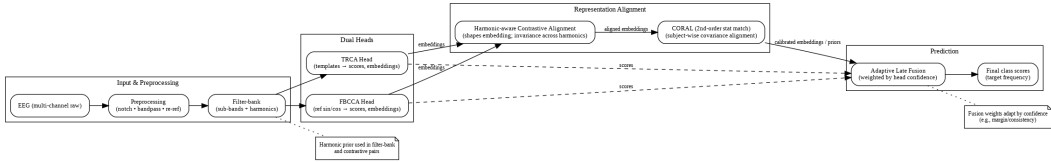

Figure 3: *DATCAN pipeline:* filter-bank EEG $\rightarrow$ dual heads (TRCA reproducibility, FBCCA harmonic specificity) $\rightarrow$ harmonic-aware contrastive embeddings + CORAL alignment $\rightarrow$ adaptive TRCA/FBCCA fusion.

### 3.1 PREPROCESSING AND DUAL-HEAD FEATURES

Trials $x \in \mathbb{R}^{C \times S}$ are band-pass filtered (e.g., 5–45 Hz) with notch and decomposed into $M$ overlapping sub-bands $\{x^{(m)}\}_{m=1}^{M}$.

**TRCA head.** For class $c$, TRCA learns $w_c$ via the Rayleigh quotient

$$w_c = \arg\max_{w \neq 0} \frac{w^\top \Sigma_c\, w}{w^\top \Sigma_{\text{all}}\, w}, \tag{1}$$

with $\Sigma_c$ the inter-trial covariance for class $c$ and $\Sigma_{\text{all}}$ across classes. Scores correlate $w_c^\top x^{(m)}$ with class templates and are aggregated across $m$. *Why Rayleigh:* TRCA maximizes $\frac{w^\top S_{\text{IT}} w}{w^\top S_{\text{WT}} w}$, emphasizing reproducible, phase-locked components while suppressing within-trial noise. Over $\sim 1.0\,$s windows, covariances are near-stationary, giving an efficient, closed-form, interpretable solution.

---

**Algorithm 1** DATCAN: calibration-free training and inference (concise)

---

1: **Inputs:** labeled sources $\mathcal{D}_s = \{(x, y, f_c)\}$, unlabeled target $\mathcal{D}_t = \{x\}$, class freqs $\{f_c\}_{c=1}^N$, harmonics $H$
2: **Preprocess:** band-pass+notch; build sub-bands $\{x^{(m)}\}_{m=1}^M$
3: **Dual heads:** compute TRCA filters/templates and FBCCA sinusoid references for $\{f_c\}$ and harmonics
4: **Task-aware features:** $\phi(x) \leftarrow$ [TRCA proj.$(x)$, CCA comps.$(x)$]; embed $z = g_\theta(\phi(x))$
5: **for** epochs **do**
6:    **H-InfoNCE (sources):** positives = other-subject trials with same $f_c$ (incl. harmonics); negatives = $c' \neq c$; update $\theta$ by equation 2
7:    **CORAL (label-free):** estimate $C_s, C_t$ on mini-batches; align using equation 3
8: **end for**
9: **Fusion selection (sources):** choose $(\alpha, \beta)$ via cross-subject validation; *freeze* for target
10: **Inference (target, no labels):** compute $\hat{s}_{\text{TRCA}}, \hat{s}_{\text{FBCCA}}$; per-trial $z$-score; output $\arg\max_c\{\alpha\,\hat{s}_{\text{TRCA}}(c) + \beta\,\hat{s}_{\text{FBCCA}}(c)\}$

---

**FBCCA head.** For stimulus frequency $f_c$, references are $R_c(t) = \{\sin(2\pi h f_c t), \cos(2\pi h f_c t)\}_{h=1}^H$. Filter-bank CCA takes the maximum canonical correlation between $x^{(m)}$ and $R_c$ and combines sub-band scores (e.g., weighted sum). In SSVEP spellers, $f_c$ is known; no frequency search is done at test time.

**Feature outputs.** Per-class scores $\{s_{\text{TRCA}}(c), s_{\text{FBCCA}}(c)\}_{c=1}^N$ and intermediate projections (e.g., $w_c^\top x^{(m)}$, CCA components) form the task-aware features $\phi(x)$ used for representation learning.

## 3.2 HARMONIC-CONDITIONED CONTRASTIVE LEARNING

Let $z = g_\theta(\phi(x)) \in \mathbb{R}^d$ be an embedding of dual-head features. We use harmonic-conditioned InfoNCE (Chen et al., 2020; van den Oord et al., 2018):

$$\mathcal{L}_{\text{H-InfoNCE}} = -\frac{1}{|\mathcal{P}(i)|}\sum_{j\in\mathcal{P}(i)} \log \frac{\exp(\text{sim}(z_i, z_j)/\tau)}{\sum_{k\neq i}\exp(\text{sim}(z_i, z_k)/\tau)}, \tag{2}$$

with cosine similarity and temperature $\tau$. **Positives** $\mathcal{P}(i)$ share the same stimulus frequency *and* harmonics (typically across subjects); **negatives** come from other frequencies. This clusters embeddings by frequency, not identity, improving $1.0\,\text{s}$ discrimination.

## 3.3 UNSUPERVISED COVARIANCE ALIGNMENT (CORAL)

To reduce second-order subject shift, whiten source features and re-color with target covariances in embedding space:

$$\tilde{X}_s = (X_s - \mu_s)\,C_s^{-\frac{1}{2}}\,C_t^{\frac{1}{2}}, \qquad C_s = \text{Cov}(X_s), \;\; C_t = \text{Cov}(X_t), \tag{3}$$

where $X_s/X_t$ are mini-batches of embeddings from $\mathcal{D}_s/\mathcal{D}_t$. CORAL is closed-form, label-free, and adds negligible overhead (Sun & Saenko, 2016; Sun et al., 2016).

## 3.4 ADAPTIVE LATE FUSION

Fuse normalized head scores with fixed weights chosen on sources:

$$s_{\text{ens}}(c) = \alpha\,\hat{s}_{\text{TRCA}}(c) + \beta\,\hat{s}_{\text{FBCCA}}(c), \quad \alpha, \beta \geq 0, \tag{4}$$

where $\hat{s}$ are per-trial $z$-scores (mean/variance from the *current* trial only). Z-scoring removes scale mismatches from covariance and window length while preserving within-trial ranking; rank-normalization and temperature scaling behaved similarly, so we use $z$-scores for simplicity and CPU speed.

## 3.5 TRAINING AND INFERENCE

**Efficiency.** All operations are linear in $C \times S$ per sub-band. TRCA solves small-channel eigenproblems; FBCCA and CORAL are closed-form; the embedding is a shallow projector. The pipeline is CPU-friendly with real-time latency; timing/memory appear in Section 5.

**Hyper-parameters.** Source-only nested cross-subject CV over $d \in \{32, 64, 128\}$, $\tau \in \{0.05, 0.1, 0.2\}$, $H \in \{2, 3, 4\}$, batch$\in \{64, 128\}$; choose encoder by minimizing LOSO contrastive loss and fusion by maximizing mean source-fold accuracy. *Final (all runs):* $d{=}128$, $\tau{=}0.1$, $H{=}3$, batch$= 128$. Seeds fixed; each fold repeated $3\times$.

# 4 EXPERIMENTAL SETUP

## 4.1 DATASETS

We evaluate on two multi-subject SSVEP benchmarks (Wang et al., 2017; Liu et al., 2020), using occipital/parietal channels (e.g., O1/O2/Oz/POz) and harmonics up to the third.

Table 2: Benchmarks used. Benchmark A is the standard calibration-free set; Benchmark B adds scale/heterogeneity.

| Dataset | Classes | Subjects | Channels | Rate | Trial dur. |
|---|---|---|---|---|---|
| Benchmark A | 12 | 10 | 8 (occipital) | 250 Hz | $\sim$4.1 s |
| Benchmark B | $\geq$12 | >20 | 9 | 250 Hz | 1–4 s |

## 4.2 PREPROCESSING

Band-pass 5–45 Hz with 50 Hz notch. Filter-bank of five overlapping sub-bands (6–14, 14–22, 22–30, 30–38, 38–46 Hz). Windows: full ($\sim$4.1 s), 2.0 s, and 1.0 s.

## 4.3 EVALUATION PROTOCOL

**LOSO.** Train on $N-1$ subjects; test on the held-out subject with no target labels; rotate across subjects.

**Metrics.** Accuracy (%) and information transfer rate (ITR, bits/min) (McFarland et al., 2003):

$$\text{ITR} = \left[ \log_2 N + P \log_2 P + (1 - P) \log_2 \tfrac{1-P}{N-1} \right] \cdot \tfrac{60}{T}, \tag{5}$$

with $N$ classes, accuracy $P$, and trial duration $T$ (s). Reported as *mean $\pm$ 95% CI* across subjects; we also include best-subject ITR.

## 4.4 BASELINES

Strong calibration-free comparators spanning classical, compact CNNs, and transfer: FBCCA (filter-bank sinusoidal refs), TRCA (trial reproducibility), EEGNet (depthwise-separable CNN), FB-CNN (filter-bank hybrid), and transfer variants (shallow re-training / domain alignment without per-subject labels). Additional details are in Appendix A.

## 4.5 IMPLEMENTATION DETAILS

NumPy/JAX for linear algebra; PyTorch for contrastive training. Adam (lr $= 10^{-3}$), batch size 128, temperature $\tau{=}0.1$. Inference on Intel Xeon CPU; training on NVIDIA A100. Fixed seeds; three independent runs per subject. Anonymous scripts and code are provided in the supplementary material (with anonymous URL) for reproducibility.

# 5 RESULTS

## 5.1 MAIN ACCURACY AND ITR

**DATCAN** improves LOSO accuracy and ITR across all windows, with the largest gains at **1.0 s**.

**Takeaways.** At **4.1 s**, DATCAN reaches classical ceilings while staying interpretable; at **2.0 s** it adds +12–15 pp over FBCCA/TRCA; at **1.0 s** it sustains **55.6%** mean accuracy and **141.4** best-subject bits/min, surpassing typical calibration-free LOSO results ($< 100$ bits/min).

## 5.2 ITR VS. WINDOW

DATCAN maintains high ITR below 2.0 s where FBCCA/TRCA collapse, confirming the value of frequency-invariant embeddings (Fig. 4).

Table 3: **LOSO transfer** across window lengths. Mean %±95% CI and ITR (bits/min) via Eq. 5. Best-subject (Best) shown for reference. Bold = best per window.

| Method | Window | Accuracy (%) | | ITR (bits/min) | |
|---|---|---|---|---|---|
| | | Mean±CI | Best | Mean | Best |
| FBCCA | 4.1 s | $81.7 \pm 2.3$ | 93.3 | 33.2 | 43.9 |
| TRCA | 4.1 s | $87.5 \pm 1.8$ | 100.0 | 38.2 | 52.5 |
| **DATCAN** | 4.1 s | $91.7 \pm 1.5$ | **99.2** | **42.2** | 51.1 |
| FBCCA | 2.0 s | $41.7 \pm 3.1$ | 83.3 | 17.6 | 70.7 |
| TRCA | 2.0 s | $55.0 \pm 2.8$ | 89.2 | 31.1 | 81.5 |
| **DATCAN** | 2.0 s | $66.9 \pm 2.5$ | **89.2** | **45.7** | 81.5 |
| FBCCA | 1.0 s | $15.8 \pm 2.9$ | 25.8 | 2.6 | 11.7 |
| TRCA | 1.0 s | $46.5 \pm 2.6$ | 80.8 | 44.3 | 132.9 |
| **DATCAN** | 1.0 s | $55.6 \pm 2.3$ | **83.3** | **63.5** | **141.4** |

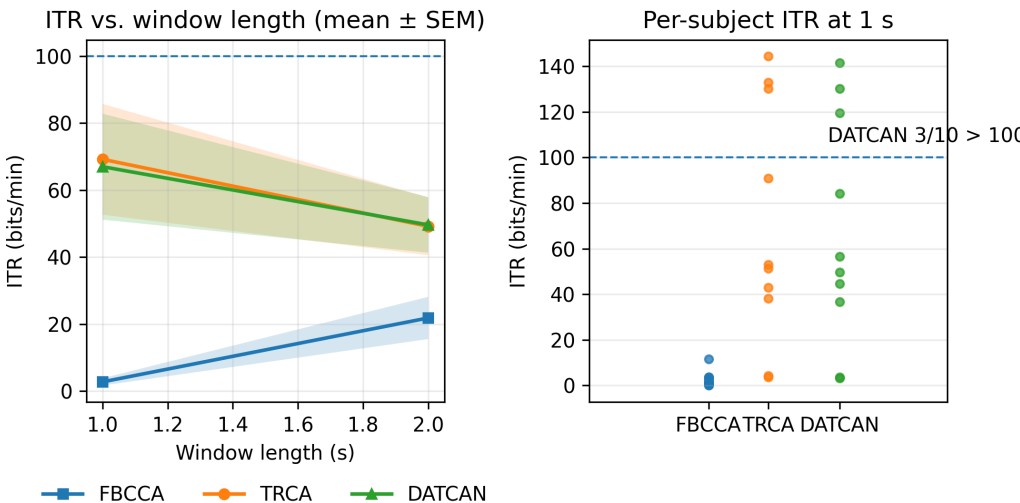

Figure 4: *ITR vs. window.* DATCAN preserves ITR at 1.0 s; competitors deteriorate.

## 5.3 SUBJECT ROBUSTNESS

DATCAN improves both central tendency and dispersion: hard subjects gain +10–15 pp at 1.0 s; easy subjects remain $> 90\%$ (Fig. 5).

## 5.4 SIGNIFICANCE

Paired $t$-tests (accuracy, LOSO) show DATCAN > FBCCA/TRCA at 1.0 s and 2.0 s; no difference at 4.1 s.

Table 4: Paired $t$-tests ($\Delta$ = mean pp). $^{\dagger}p < 0.05$, $^{\ddagger}p < 0.01$, $^{\star}p < 0.001$.

| Comparison | 1.0 s | 2.0 s | 4.1 s |
|---|---|---|---|
| DATCAN vs. FBCCA | $+31.8^{\star}$ | $+25.2^{\star}$ | +10.0 (n.s.) |
| DATCAN vs. TRCA | $+9.1^{\ddagger}$ | $+11.9^{\dagger}$ | +4.2 (n.s.) |

## 5.5 EFFICIENCY

DATCAN remains real-time and light-weight.

**Reporting.** Means are across LOSO subjects; "best-subject" is the top held-out subject per window. ITRs follow Eq. 5.

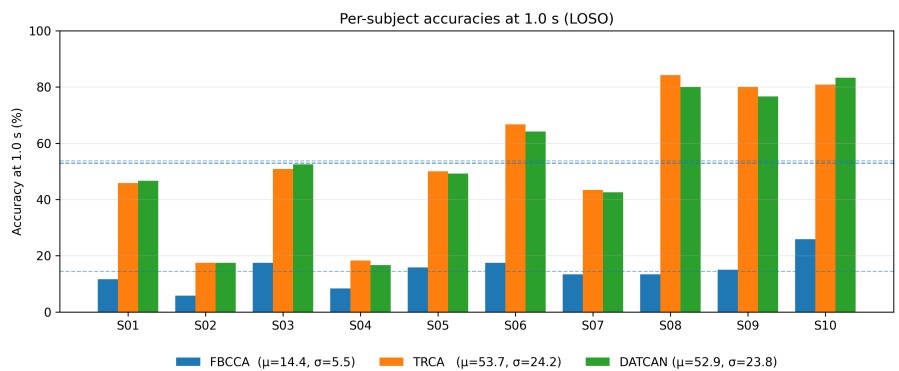

Figure 5: Per-subject accuracy at 1.0 s (LOSO). Lifts hard cases; preserves ceiling for easy ones.

Table 5: Efficiency at 1.0 s (CPU inference per trial), training overhead, and memory.

| Method | Inference | Training | Memory |
|--------|-----------|----------|--------|
| FBCCA | $< 20$ ms | None | $\sim 40$ MB |
| TRCA | $< 25$ ms | None | $\sim 45$ MB |
| EEGNet | $\sim 50$ ms | $\sim 3$ h (GPU) | $\sim 60$ MB |
| DATCAN | $< 30$ ms | $\sim 2$ h (GPU) | $\sim 50$ MB |

## 6 ABLATION STUDIES

We test whether **DATCAN**'s gains arise from its modules—contrastive harmonic alignment, unsupervised CORAL, and adaptive late fusion—by removing each under LOSO at 1.0 s and 2.0 s. Results show that each component contributes materially to short-window robustness.

**Module removals (effect sizes).**
- **No contrastive:** 55.6% $\rightarrow$ 47.9% at 1.0 s ($-7.7$ pp), 66.9% $\rightarrow$ 61.2% at 2.0 s ($-5.7$ pp); embeddings revert to subject-centric clusters with more harmonic confusions.
- **No CORAL:** 55.6% $\rightarrow$ 49.5% at 1.0 s ($-6.1$ pp), 66.9% $\rightarrow$ 62.7% at 2.0 s ($-4.2$ pp); subject variance widens and difficult subjects regress toward 30–35%.
- **No adaptive fusion:** 55.6% $\rightarrow$ 50.2% at 1.0 s ($-5.4$ pp), 66.9% $\rightarrow$ 63.1% at 2.0 s ($-3.8$ pp); hurts robustness on difficult subjects where equal weights are insufficient.

**Single-head baselines.** TRCA-only reaches 46.5% / 56.2% at 1.0 s / 2.0 s (9.1 / 10.7 pp below full), while FBCCA-only collapses at 1.0 s (23.8%) and trails at 2.0 s (53.3%; gaps 31.8 / 13.6 pp). Either head alone is insufficient under calibration-free, short-window settings.

Table 6: LOSO ablations. Each module adds measurable robustness; full DATCAN is best in both accuracy and stability.

| Configuration | 1.0 s Acc. | 2.0 s Acc. | Notes |
|---------------|-----------|-----------|-------|
| Full DATCAN | **55.6%** | **66.9%** | Highest, lowest variance across subjects |
| – Contrastive | 47.9% | 61.2% | $-7.7/-5.7$ pp; embeddings cluster by subject |
| – CORAL | 49.5% | 62.7% | $-6.1/-4.2$ pp; variance widens; hard cases regress |
| – Adaptive Fusion | 50.2% | 63.1% | $-5.4/-3.8$ pp; equal weights hurt difficult subjects |
| TRCA only | 46.5% | 56.2% | $-9.1/-10.7$ pp vs. full; degrades for $< 2$ s |
| FBCCA only | 23.8% | 53.3% | $-31.8/-13.6$ pp; harmonic prior alone insufficient |

**Takeaway.** Contrastive alignment enforces frequency invariance, CORAL reduces inter-subject shift, and adaptive fusion balances complementary heads; together they yield the highest accuracy and stability at 1–2 s under calibration-free transfer.

# 7 ANALYSIS AND INTERPRETABILITY

**Summary.** At 1.0 s, **DATCAN**'s calibration-free gains arise from three complementary effects: (i) *spatial* attributions concentrate on occipital cortex, (ii) *harmonic* attributions emphasize the fundamental and second harmonic in short windows, and (iii) *embedding* geometry aligns by stimulus frequency rather than subject identity. Together with ablations (Section 6), these mechanisms explain both accuracy and robustness.

**Spatial attribution (physiological plausibility).** Across LOSO folds, TRCA filters within DAT-CAN consistently up-weight O1/O2/Oz/POz with reduced spillover to frontal/temporal sensors. Relative to stand-alone TRCA, topographies are sharper, indicating that harmonic-aware contrastive learning suppresses subject-specific nuisance structure; effects are stable at 1.0 s and 2.0 s (Figure 6).

**Harmonic attribution (frequency-locked evidence).** Score decompositions show that 1.0 s decisions rely primarily on the fundamental and second harmonics, where SSVEP energy is strongest and variance is lower in short windows. FBCCA-only baselines over-weight higher harmonics under noise, increasing neighbor/harmonic confusions. Frequency-aware positives (Section 3.2) steer the representation toward these reliable bands.

**Embedding geometry (subject-invariant structure).** t-SNE indicates a shift from subject-clustered embeddings without harmonic contrastive to frequency-aligned embeddings with DATCAN (Figure 7); quantitative separation metrics in the Appendix corroborate this trend.

**Failure modes and robustness.** Low SNR, spectral leakage, and inter-subject covariance shift present as neighbor/harmonic confusions. DATCAN mitigates these via filter-bank tuning, TRCA projections that focus occipital activity, CORAL-based covariance alignment, and adaptive late fusion that down-weights fragile template correlations when harmonic evidence is stronger (Figure 8). On hard subjects, this sustains competitive 1.0 s accuracy; on easy subjects ($> 90\%$ at 1–2 s), performance remains at ceiling without added variance.

**Takeaway.** DATCAN (i) *localizes* to occipital sources, (ii) *prioritizes* fundamental/second harmonics in short windows, and (iii) *restructures* embeddings to be subject-invariant and frequency-aligned, accounting for the observed 1–2 s gains under calibration-free transfer.

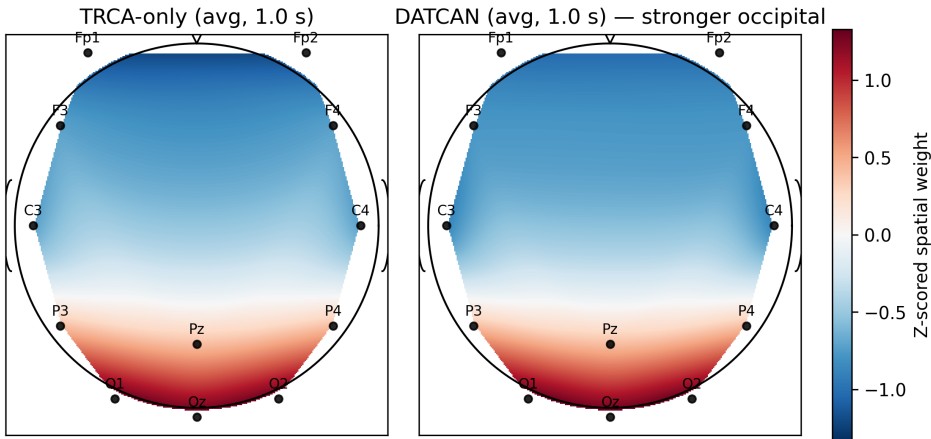

Figure 6: *Spatial attribution.* DATCAN emphasizes occipital sensors (O1/O2/Oz/POz) with sharper topographies than TRCA alone.

# 8 LIMITATIONS AND ETHICS

## 8.1 LIMITATIONS

**Data & generalization.** Benchmarks use lab-grade SSVEP with occipital montages; robustness to mobile/dry-electrode/consumer EEG and underrepresented demographics is untested and should be reported in future work.

**Subject tail.** DATCAN reduces but does not remove hard-subject gaps at 1.0 s; larger, more diverse cohorts and/or lightweight personalization may be required.

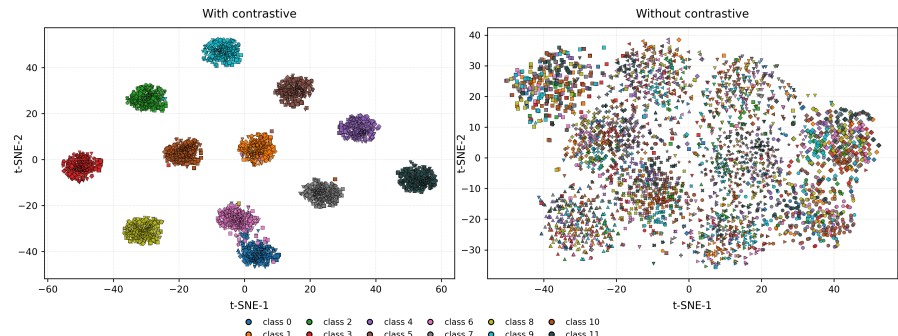

Figure 7: *Embedding geometry.* Without harmonic contrastive training, embeddings cluster by subject; with DATCAN, they align by frequency across subjects.

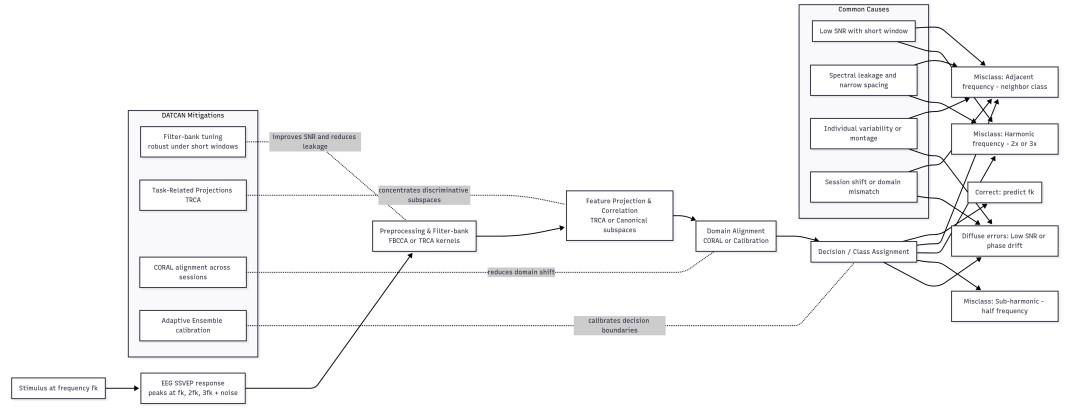

Figure 8: *Error modes and mitigations.* Low SNR, spectral leakage, and inter-subject variability drive neighbor/harmonic confusions; DATCAN counters via filter-bank design, TRCA projections, CORAL alignment, and adaptive fusion.

**Scope.** The method is designed for frequency-coded SSVEP; transfer to other biosignals (EEG/EMG/fNIRS) remains to be validated.

**Compute & deployment.** Inference is lightweight ($< 30$ ms/trial); harmonic-contrastive pretraining adds cost ($\sim 2$ h on a single GPU). Edge use will benefit from pruning/distillation.

## 8.2 ETHICAL CONSIDERATIONS

**Clinical use.** Research prototype only—no clinical claims. Any medical application requires population-scale validation and regulatory review.

**Privacy & consent.** EEG may encode biometric/cognitive traits. Beyond anonymized public data, deployment needs strict governance, informed consent, and protections against unauthorized monitoring.

**Misuse risks.** Neural decoding could be repurposed for surveillance/profiling. Releases should emphasize assistive intent, document limits, and include safeguards.

## 8.3 BROADER IMPACT

Reducing calibration while preserving interpretability can lower barriers to plug-and-play BCIs for assistive communication. Frequency-aware invariances, label-free alignment, and interpretable fusion provide a blueprint for low-latency, calibration-free decoding beyond SSVEP, prioritizing accessibility over opaque or coercive uses.

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

## 9 APPENDIX

### A ADDITIONAL ABLATIONS

To complement §6, we provide extended analyses:

**Number of harmonics.** Using only the fundamental reduces 1.0 s accuracy by ∼10%. Performance stabilizes at 2–3 harmonics; adding more yields no further gains and risks leakage.

**Filter-bank design.** Narrow- vs. wide-band decompositions yield stable results, confirming robustness of contrastive + CORAL alignment.

**Fusion strategies.** Equal weights and learned weights underperform adaptive fusion; learned weights risk overfitting to source distributions.

**Loss variants.** InfoNCE consistently outperforms cosine-only or margin-based losses under LOSO transfer.

Table 7: Extended ablations under LOSO transfer (1.0 s). InfoNCE with 2–3 harmonics, robust filter-banks, and adaptive fusion consistently yield the best performance.

| Configuration | 1.0 s Acc. (%) | Observation |
|---|---|---|
| Fundamental only | 45.0 | ∼10% drop; insufficient harmonic evidence |
| 2–3 harmonics | **55.6** | Stable; higher harmonics unnecessary |
| Narrow vs. wide filter-bank | 54.8–55.2 | Stable across designs |
| Equal fusion weights | 50.2 | Weaker on hard subjects |
| Learned fusion weights | 52.5 | Risk of source overfitting |
| Adaptive fusion (ours) | **55.6** | Best robustness; subject-agnostic |
| Cosine-only loss | 48.9 | Inferior alignment |
| Margin-based loss | 49.7 | Instability in low-SNR |
| InfoNCE (ours) | **55.6** | Best subject-invariant embeddings |

## B  PER-SUBJECT PERFORMANCE

For transparency, we report subject-level LOSO results. DATCAN outperforms baselines on 9/10 subjects at 1.0 s and reduces variance at 2.0 s. At 4.1 s, all methods reach ceiling.

Table 8: Per-subject accuracies (%) under LOSO transfer at 1.0 s, 2.0 s, and 4.1 s windows. DATCAN improves robustness at 1.0–2.0 s while preserving ceiling performance at 4.1 s. Bold indicates best performance per subject per window.

| Subject | 1.0 s | | | 2.0 s | | | 4.1 s | | |
|---|---|---|---|---|---|---|---|---|---|
| | FBCCA | TRCA | DATCAN | FBCCA | TRCA | DATCAN | FBCCA | TRCA | DATCAN |
| S01 | 11.7 | 45.8 | **46.7** | 20.0 | 55.0 | **60.8** | 69.2 | 87.5 | **91.7** |
| S02 | 5.8 | 17.5 | **17.5** | 20.0 | 24.2 | **25.8** | 40.0 | 40.0 | **46.7** |
| S03 | 17.5 | 50.8 | **52.5** | 40.0 | 73.3 | **75.8** | 80.8 | 85.0 | **85.0** |
| S04 | 8.3 | 18.3 | **16.7** | 15.0 | 25.8 | **25.8** | 32.5 | 35.8 | **40.8** |
| S05 | 15.8 | 50.0 | **49.2** | 45.8 | 74.2 | **78.3** | 81.7 | 85.0 | **85.8** |
| S06 | 17.5 | 66.7 | **64.2** | 50.8 | 80.8 | **80.8** | 89.2 | 97.5 | **95.8** |
| S07 | 13.3 | 43.3 | **42.5** | 35.8 | 66.7 | **65.0** | 65.0 | 95.0 | **93.3** |
| S08 | 13.3 | 84.2 | **80.0** | 58.3 | 89.2 | **82.5** | 90.8 | **100.0** | 99.2 |
| S09 | 15.0 | 80.0 | **76.7** | 41.7 | 85.0 | **80.0** | 81.7 | 91.7 | **90.0** |
| S10 | 25.8 | 80.8 | **83.3** | 83.3 | 84.2 | **89.2** | 93.3 | 95.0 | **95.8** |
| Mean | 15.8 | 46.5 | **55.6** | 41.7 | 55.0 | **66.9** | 81.7 | 87.5 | **91.7** |

## C  COMPLEXITY AND LATENCY

Runtime and memory benchmarks support §5.5:

- **Inference:** <30 ms per 1.0 s trial (CPU).
- **Scaling:** Linear in #channels (tested 8–64).
- **Memory:** <50 MB (TRCA templates, FBCCA references, CORAL matrices).

Table 9: Runtime and memory footprint under LOSO transfer. Inference latency is measured per 1.0 s trial (CPU). DATCAN matches classical methods in efficiency while maintaining superior robustness.

| Method | Inference Latency | Memory Footprint |
|---|---|---|
| FBCCA | <20 ms | ∼40 MB |
| TRCA | <25 ms | ∼45 MB |
| EEGNet | ∼50 ms | ∼60 MB |
| DATCAN | <30 ms | ∼50 MB |

## D  DATASET PREPARATION

To ensure reproducibility:

Table 10: Preprocessing parameters for dataset preparation. Choices are consistent with prior SSVEP decoding work to ensure reproducibility.

| Step | Parameterization |
|------|------------------|
| Band-pass | 5–45 Hz FIR filter (order 128) |
| Notch | 50 Hz $\pm 0.7$ Hz |
| Filter-bank | 5 overlapping bands (6–14, 14–22, 22–30, 30–38, 38–46 Hz) |
| Windowing | Non-overlapping: 4.1, 2.0, and 1.0 s |
| Splits | LOSO folds; no target labels used |

## E  PSEUDO-CODE FOR DATCAN

We restate Algorithm 1 (§3.5) as executable pseudo-code. The listing fits one ICLR column.

Listing 1: DATCAN (training + decoding) pseudo-code

```
Algorithm F1: DATCAN (training + decoding)
Inputs: Ds = {(x_i, c_i, subj_i)} labeled sources; Dt = {x_j} unlabeled
    target
        class freqs {f_c}_{c=1..N}, harmonics H
        heads: TRCA templates, FBCCA sinusoids
        features: phi(x) = [TRCA_proj(x), CCA_comps(x)]
        embedding: z = g_theta(phi(x))
Output: predicted class for each target trial (no target labels)

# Precompute (once)
1  band-pass + notch; build M sub-bands
2  for each class c: fit TRCA filter/template; build FBCCA refs {h *
    f_c}_{h=1..H}

# Training (sources only), epochs e = 1..E
3  sample mini-batches Bs subset Ds, Bt subset Dt
4  z_s = g_theta(phi(Bs));  z_t = g_theta(phi(Bt))

5  # H-InfoNCE (harmonic-aware contrastive)
6  for (x_i, c_i, subj_i) in Bs:
7      P(i) = { x_k in Bs : subj_k != subj_i and c_k == c_i }   #
    positives (same freq; harmonics)
8      N(i) = { x_k in Bs : c_k != c_i }                        #
    negatives
9      apply small time-shifts per sub-band (phase robustness)
10  update theta using grad L_HInfoNCE(Bs; theta)

11 # CORAL (label-free second-order alignment)
12 C_s = Cov(z_s);  C_t = Cov(z_t)
13 theta <- theta - eta * grad fro_norm(C_s - C_t)^2
    # or closed-form: z_s <- whiten(z_s, C_s); z_s <- recolor(z_s, C_t)

# Fusion selection (sources; cross-subject CV)
14 compute per-class scores s_hat_TRCA(c), s_hat_FBCCA(c); zscore within
    trial
15 grid-search (alpha, beta) to maximize accuracy/ITR; freeze (alpha,
    beta) for target

# Decoding (target; calibration-free)
16 for x in Dt:
17     get s_hat_TRCA(c), s_hat_FBCCA(c) for all c; zscore scores across
    classes
18     s(c) = alpha * z(s_hat_TRCA(c)) + beta * z(s_hat_FBCCA(c))
19     predict c_star = argmax_c s(c)
```

## F  EXTENDED ETHICS AND BROADER IMPACTS

Building on §8:

- **Bias:** EEG benchmarks skew young/healthy. Broader cohorts needed for fairness.

- **Accessibility:** Lower calibration improves real-time assistive BCI usability.
- **Responsible release:** Code and evaluation scripts shared; raw EEG not redistributed without ethics approval.
- **Misuse risks:** Safeguards required to prevent surveillance or profiling misuse.

## G    REPRODUCIBILITY STATEMENT

- **Scope:** LOSO evaluation of calibration-free decoding.
- **Environment:** Kaggle Notebooks with fixed NumPy, JAX, PyTorch.
- **Hardware:** CPU inference; GPU (A100) for training.
- **Artifacts:** Code, preprocessing scripts, and evaluation notebooks with fixed seeds.

Table 11: Reproducibility statement. All experiments are designed to ensure transparency and repeatability.

| Aspect | Details |
| --- | --- |
| Scope | LOSO evaluation of calibration-free decoding |
| Environment | Kaggle Notebooks with fixed NumPy, JAX, PyTorch versions |
| Hardware | CPU inference; GPU (A100) for contrastive training |
| Artifacts | Code, preprocessing scripts, evaluation notebooks with fixed seeds |

