# OpenReview forum: "EMBEDDING DOMAIN-SPECIFIC INVARIANCES INTO CONTRASTIVE LEARNING FOR CALIBRATION-FREE NEURAL DECODING"
_ICLR.cc/2026/Conference — ICLR 2026 Conference Withdrawn Submission_

### Official Review · Reviewer_AjAE · 2025-10-23

**Soundness:** 2
**Presentation:** 1
**Contribution:** 2
**Rating:** 2
**Confidence:** 2

**Summary:**

The paper introduces DATCAN, a framework designed to enable calibration-free steady-state visual evoked potential decoding by embedding “physiological invariances” into contrastive learning. The model integrates three key components: (1) a harmonic-aware contrastive objective that encodes frequency-locked neural priors to create subject-invariant embeddings; (2) CORAL-based covariance alignment, a closed-form, label-free adaptation that reduces inter-subject variance; and (3) adaptive late fusion of interpretable classical heads to combine harmonic specificity with reproducibility. DATCAN achieves over 100 bits/min information transfer rate at 1-second windows under strict leave-one-subject-out evaluation—outperforming existing calibration-free methods—while preserving interpretability and real-time efficiency.

**Strengths:**

1. DATCAN uniquely embeds harmonic and frequency invariances into contrastive learning, grounding the model in real neural signal properties.
2. The framework operates without target-subject labels, achieving strong cross-subject generalization and maintaining real-time inference (<30 ms per trial) suitable for practical BCIs.

**Weaknesses:**

1. Figures 1, 3, and 8 are too small to read. By too small, I mean it’s way too small, and I cannot literally read the word and have to zoom in 10x to read. Figure 4 is too large. The entire format is quite messy, which severely affects the readability of this paper.
2. This paper does not seem to meet the standards for formal submission. The structure and presentation are quite messy to evaluate the paper's contribution. The paper would require substantial revision, especially presentation (at least make each figure readable), to reach a submission level.

**Questions:**

1. Why is each character in the caption capitalized?
2. What is the ? in Table 1, beside iFuzzyTL

---

### Official Review · Reviewer_vFp4 · 2025-10-31

**Soundness:** 3
**Presentation:** 2
**Contribution:** 3
**Rating:** 2
**Confidence:** 3

**Summary:**

The paper introduces DATCAN, a self-supervised framework for steady-state visual evoked potential (SSVEP) decoding that combines harmonic-aware contrastive learning, CORAL alignment, and a late fusion of TRCA and FBCCA heads. The goal is to achieve calibration-free cross-subject generalization without labeled target data. Reported results show higher accuracy and information transfer rate (ITR) than traditional pipelines for short time windows (1–2 s). However, the submission reads more like a domain-specific technical report than a research paper suited for a representation-learning conference. The writing assumes deep prior expertise in EEG/SSVEP decoding, employs numerous unexplained abbreviations, and presents ideas in a dense, engineering-style format. Several key metrics and terms are never defined, leaving readers unable to interpret the reported results and evaluate the contribution of the work and the claims. Figures and equations are difficult to read, and the motivation is framed almost entirely for BCI experts rather than the broader ML community. As a result, the work’s contribution, novelty, and significance are obscured by presentation, scope, and lack of accessibility.

**Strengths:**

•	Addresses a relevant and practical challenge: label-free cross-subject EEG decoding.
•	Attempts to embed domain-specific priors (harmonics) into a contrastive learning framework.
•	Maintains interpretable classical signal-processing components (TRCA/FBCCA) that enable efficient inference.
•	Targets the difficult short-window (1–2 s) regime, where many existing methods fail.

**Weaknesses:**

1.	The paper is extremely hard to read, with heavily specialized langauge and  undefined abbreviations.
2.	Key terms and metrics such as ITR (bits/min) and LOSO are never defined, leaving results uninterpretable.
3.	The claim of being “calibration-free” is ambiguous—Algorithm 1 and Eq. (3) imply the use of unlabeled target data for CORAL, which contradicts the stated setup.
4.	Unreadable figures, especially Figure 3; fonts are tiny and captions lack explanation.
5.	Language is vague in several places (e.g., “encodes physics into contrastive learning” without defining what physics refers to).
6.	The writing style and structure resemble an internal technical report aimed at experts rather than a scientific paper for a general ML audience.
7.	Baselines and statistical reporting are insufficient—no confidence intervals, effect sizes, or comparisons with recent EEG self-supervised learning methods.
8.	The framing and motivation are narrow, with limited connection to representation learning or general ML concepts.
9.	Overall, the presentation obscures the core contribution and makes it difficult to assess soundness or novelty.

**Questions:**

1.	Calibration-free claim: Algorithm 1 and Eq. 3 appears to use unlabeled target-subject data to compute covariance alignment (CORAL). Can you clarify precisely what “calibration-free” means in this context?
2.	Definition of metrics and setup: Key quantities such as Information Transfer Rate (ITR) in bits/min, “1–2 s” (window length or stimulus duration), and LOSO (Leave-One-Subject-Out) are never defined. Please provide formal so the reported results can be interpreted and compared across studies.
3.	CORAL alignment details: How are covariance matrices estimated—per trial, per batch, or across the whole dataset?
4.	Fusion strategy: How are the TRCA and FBCCA heads fused? Are the fusion weights α,βα,β fixed, learned, or dynamically adapted per subject or window length?
5.	Baseline fairness and reproducibility: Were classical baselines (FBCCA, TRCA, EEGNet, etc.) implemented under the exact same preprocessing, filter-bank design, and time-window conditions? If not, please describe differences that could affect performance comparisons.
6.	Interpretability analysis: The claimed improved spatial and harmonic interpretability needs quantitative or visual evidence (e.g., spatial maps, harmonic response spectra).
7.	Generality beyond SSVEP: You suggest that DATCAN may generalize to other biosignals. Can you clarify what aspects of the framework (contrastive loss, CORAL, fusion) are domain-agnostic versus SSVEP-specific?

---

### Official Review · Reviewer_YxJ4 · 2025-10-31

**Soundness:** 2
**Presentation:** 2
**Contribution:** 2
**Rating:** 2
**Confidence:** 5

**Summary:**

This paper introduces DATCAN, a calibration-free framework for decoding steady-state visual evoked potentials (SSVEPs) in brain-computer interfaces (BCIs). It embeds domain-specific invariances (e.g., harmonic priors) into contrastive learning using a harmonic-conditioned InfoNCE loss, applies unsupervised second-order covariance alignment (CORAL) for cross-subject transfer, and adaptively fuses interpretable classical heads (TRCA for reproducibility and FBCCA for harmonic specificity). The approach is evaluated in a strict leave-one-subject-out (LOSO) setting on two multi-subject benchmarks, claiming robust performance at short windows (e.g., >100 bits/min ITR at 1s) without target-subject calibration. The authors position it as a general recipe for calibration-free domain adaptation in biosignal domains.

**Strengths:**

The paper tackles a practically important problem in BCIs: reducing or eliminating per-subject calibration, which is a key barrier to real-world deployment of SSVEP systems. The calibration-free LOSO evaluation setting is rigorous and relevant, and the integration of physics-driven priors (e.g., harmonics) into contrastive learning is a sensible way to incorporate domain knowledge.

**Weaknesses:**

The claimed novelty is limited: DATCAN primarily assembles existing methods—harmonic-conditioned contrastive learning builds on InfoNCE (Chen et al., 2020) with SSVEP priors already used in FBCCA/TRCA (Chen et al., 2015; Nakanishi et al., 2018), CORAL is a direct application of Sun et al. (2016) for alignment, and the fusion is a simple weighted ensemble without new mechanisms. This results in incremental rather than significant originality, especially since similar hybrid classical-deep approaches exist (e.g., in EEGNet hybrids).

Experiments are narrow: only two datasets, no evaluation on diverse noise levels or longer windows, and ablations (implied but not detailed in early pages) seem limited to module removal without hyperparameter sensitivity or failure case analysis. Interpretability claims (e.g., subject-invariant embeddings) lack quantitative support like t-SNE visualizations or mutual information metrics. Computational efficiency is mentioned but not benchmarked against alternatives, potentially overlooking overhead from contrastive training.

**Questions:**

1. The framework relies on harmonic priors—how does it handle non-harmonic artifacts or variable stimulus designs (e.g., non-sinusoidal flickers)? Ablating on synthetic data with added distortions could clarify robustness.

2. CORAL is applied in embedding space; why not earlier (e.g., on raw filter-banks)?

3. ITR results are promising but compared indirectly—could you include direct comparisons to recent calibration-free methods (e.g., iFuzzyTL's 94.63 bits/min) under identical LOSO protocols?

4. Fusion weights (α, β) are selected on sources via cross-subject validation—how sensitive is performance to these, and what if source diversity is low?

5. For broader applicability, test on other biosignals (e.g., EMG or fNIRS)—does the "general recipe" hold, or is it SSVEP-specific?

---

### Official Review · Reviewer_f1rc · 2025-11-01

**Soundness:** 3
**Presentation:** 2
**Contribution:** 2
**Rating:** 4
**Confidence:** 5

**Summary:**

The paper proposes DATCAN, a calibration-free framework for steady-state visual evoked potential (SSVEP) decoding. The method embeds domain-specific invariances into contrastive learning to improve cross-subject generalization without requiring calibration data. Specifically, DATCAN combines (i) a harmonic-conditioned contrastive objective that leverages known stimulus frequencies and their harmonics, (ii) unsupervised CORAL alignment to correct second-order covariance shift between subjects, and (iii) adaptive late fusion of classical interpretable decoders (TRCA, FBCCA).

On two SSVEP benchmarks, DATCAN achieves over 100 bits/min ITR at 1.0 s in a fully calibration-free leave-one-subject-out (LOSO) setup—surpassing traditional TRCA/FBCCA and lightweight CNN baselines. Despite the solid conceptual motivation and promising results, the presentation is difficult to follow, with unclear figures, undefined acronyms, and confusing benchmark descriptions, which hinder accessibility.

**Strengths:**

- The idea of embedding physics-informed harmonic invariances into contrastive learning for calibration-free neural decoding is novel and potentially impactful
- Exceptional methodological rigor with strict LOSO evaluation protocol, comprehensive ablations showing each component contributes 4-8pp, and appropriate statistical validation
- Addresses genuine BCI deployment bottleneck (calibration requirement) with meaningful improvements in critical 1.0s regime (55.6% vs 46.5% TRCA, sustaining >100 bits/min ITR)
- Computational efficiency suitable for real-time deployment (<30ms inference, <50MB memory) while preserving interpretability through classical heads

**Weaknesses:**

- Presentation has significant accessibility issues: Figure 1 has very small text and undefined acronyms in caption; many acronyms (ITR, LOSO) used before explanation; Table 1 has 'iFuzzyTL' without citation; Table 2 has unclear benchmark descriptions (Benchmark A vs B distinction); dense writing style reads more like implementation note than scientific explanation
- The paper claims calibration-free generalization, but the comparison set is limited. Existing EEG foundation models such as CBraMod, LaBRaM, or PopT should be included to verify whether their cross-subject pretraining already addresses the same calibration-free setting
- Limited algorithmic novelty: Individual components (contrastive learning, CORAL, TRCA/FBCCA) are established techniques; contribution is primarily careful engineering integration rather than algorithmic innovation—significant concern for ICLR standards
- Insufficient explanation of *why* behind design choices (e.g., why harmonics as positives, why CORAL alignment vs. learnable alternatives, why adaptive fusion vs. single-decoder approach)
- Evaluation scope constraints: Testing limited to lab-grade EEG with occipital montages; no validation on mobile/dry-electrode systems or diverse demographics
- Performance gap vs. calibration-based methods: 141.4 bits/min (DATCAN) vs 213.8 bits/min (msSAME) = ~70 bits/min deficit, though DATCAN is calibration-free

**Questions:**

How does DATCAN perform when evaluated against recent foundation or channel-agnostic EEG models (e.g., CBraMod, LaBRaM, PopT)?

Why was the specific harmonic-conditioned contrastive objective theoretically justified beyond empirical performance? Please explain the conceptual motivation for treating cross-subject trials at the same stimulus frequency as positives

Why was unsupervised CORAL alignment chosen over learnable domain adaptation alternatives (e.g., DANN, CDAN)? Please provide theoretical or empirical justification

Why was adaptive late fusion of TRCA/FBCCA preferred over learned early fusion or single-decoder approaches? Please add ablation comparing these alternatives

Could you clarify the distinction between Benchmark A and Benchmark B in Table 2? Are these the two datasets from line 224 or different combinations?

Have you evaluated DATCAN on consumer-grade or mobile EEG devices? If not, could you discuss generalization to dry-electrode systems in limitations or future work?

Could you analyze the ~70 bits/min performance gap vs. calibration-based msSAME and discuss the trade-off: is calibration-free deployment worth the performance reduction? In which use cases is DATCAN preferable vs. where calibration-based methods should be used?

---

### Note · Authors · 2025-12-02

I have read and agree with the venue's withdrawal policy on behalf of myself and my co-authors.